# Anthropogenic Activities and the Problem of Antibiotic Resistance in Latin America: A Water Issue

**Delfina C. Domínguez** [1,*]**, Luz María Chacón** [2,*] **and D'Janique Wallace** [3,4]

1 Clinical Laboratory Science Program, Department of Public Health Sciences, The University of Texas at El Paso, 500 W. University Ave. HSSN 420, El Paso, TX 79968, USA

2 Health Research Institute (INISA), University of Costa Rica, P.O. Box 11501-2060, San Pedro de Montes de Oca 11502, Costa Rica

3 Biology Department, El Paso Community College, P.O. Box 20500, El Paso, TX 79998, USA; mookiedw1@gmail.com

4 Biological Sciences Department, The University of Texas at El Paso, 500 W University Ave., El Paso, TX 79968, USA

* Correspondence: delfina@utep.edu (D.C.D.); luz.chacon@ucr.ac.cr (L.M.C.); Tel.: +1-915-747-7238 (D.C.D.); +506-2511-2150 (L.M.C.)

**Abstract:** Antibiotics revolutionized modern medicine and have been an excellent tool to fight infections. However, their overuse and misuse in different human activities such as health care, food production and agriculture has resulted in a global antimicrobial resistance crisis. Some regions such as Latin America present a more complex scenario because of the lack of resources, systematic studies and legislation to control the use of antimicrobials, thus increasing the spread of antibiotic resistance. This review aims to summarize the state of environmental antibiotic resistance in Latin America, focusing on water resources. Three databases were searched to identify publications on antimicrobial resistance and anthropogenic activities in relation to natural and artificial water ecosystems. We found that antibiotic resistant bacteria, mainly against beta lactam antibiotics, have been reported in several Latin American countries, and that resistant bacteria as well as resistant genes can be isolated from a wide variety of aquatic environments, including drinking, surface, irrigation, sea and wastewater. It is urgent to establish policies and regulations for antibiotic use to prevent the increase of multi-drug resistant microorganisms in the environment.

**Keywords:** Latin America; antibiotic resistance; antibiotic-resistance genes; human activities; water sources; anthropogenic activities; ESBL

## 1. Introduction

Antimicrobial resistance (AMR) is considered a major global health crisis faced by humankind. To combat this serious problem various initiatives have been created worldwide, including: the Transatlantic Taskforce on Antimicrobial Resistance (TATFAR), created in 2009 by Canada, European Union, Norway and the U.S.; the Global Antibiotic Resistance Partnership (GARP) created in 2009 to develop policy proposals on AMR in Low-income (LIC) and Middle-income countries (MIC); the Global Health Security Agenda (GHSA), a group of 70 countries to address global threats; the high level meeting on AMR by the United Nations in 2016; One Health Approach, which was designed to improve public health [1–3]; and the AMR Challenge 2018–2019, a yearlong effort to fight AMR among 350 organizations across the world [4,5]. Recent data by the CDC indicate that more than 2.8 million antibiotic resistance infections and 35,000 deaths occur each year in the U.S. [6]. In China, about 100,000 deaths are due to multi-drug resistant organisms (MDRO) [3,7]. At present, it is estimated that mortality rates due to AMR are 700,000 deaths annually [8]. The same report predicts that by 2050, the number of deaths may increase to 10 million per year.

Each year, the global consumption of antibiotics is estimated to be 70 billion standard units/year for humans [9]. Antibiotic usage is projected to increase by 67% within the next

fifteen years [10]. The cost of AMR is not only monetary but also affects social activities and the workforce. The global economic impact is broad, affecting healthcare systems, healthcare providers, patients, pharmaceutical organizations and international policy [11]. The reduction of productivity can reach US $1.5 billion per year [7].

Global drivers for AMR include antibiotic overuse (humans and animals), antibiotic misuse, suboptimal therapeutic doses, inappropriate disposal of antimicrobials, and the use of antibiotics in agriculture, aquaculture and in cattle as prophylaxis and growth promoters for commercial purposes, which may contaminate rivers, streams, groundwater and soil [12–14].

The prolonged use of antimicrobials in livestock and aquaculture is of great concern, since the presence of antibiotic residues are found in the edible tissues of treated animals [15,16]. The use of antibiotics in animals varies significantly depending on the country. For example in Europe, North America and Japan, regulation of antibiotics is strict compared to developing countries where there is a lack of regulation and enforcement of the use of antibiotics [17]. However, the magnitude of the threat of AMR for public health from agriculture, livestock and aquaculture has not been quantified [15,17,18].

Dissemination of AMR occurs first through vertical transfer, bacterial mutations passed from bacterial parent cell to offspring, and second by horizontal transfer, which may include bacterial genetic elements such as plasmids and integrons or vectors such as humans and animals or vehicles such as water and food [14,19]. The mobility of antibiotic resistant genes (ARG) and the potential transmission to populations, animals and the environment is a major challenge that needs to be addressed [19]. Contamination hotspots include waste water treatment plants (WWTP), hospital sewage discharges and pharmaceutical industry waste [12,20].

AMR has affected all world countries, but the burden is disproportionally higher in low-income countries and middle-income countries. Challenges faced by Low- and Middle-income countries (LMIC) to combat AMR include political, economic, social, industrial, technological and environmental factors [21,22]. Reports from LMIC show that AMR is increasing and it is more common than in High income countries (HIC), not only in health care centers but also in the environment [19,23]. However, there is no conclusive evidence since this disparity may be due to sample bias, lack of standardized laboratory practices and underutilization of microbiological procedures [22,24].

Some of the factors favoring AMR in LMIC include population density, migration, and level of education, and infrastructure issues such as sanitation, poor hygiene, lack of potable water, untreated wastewater, illegal discharges and solid waste disposal. The low economy of these countries is the cause of lack of resources in medical centers, limited attention to prevention and infection control, and lack of microbiology laboratories and thus microbiology practices. Finally, poor governance contributes to lack of regulations in the antibiotic industry (forged antibiotics), lack of policies and overall ineffective health care centers [19,21,24,25].

In Latin America, MDRO are the leading cause of hospital acquired infections. According to surveillance data from the Latin American Network for antimicrobial Resistance Surveillance (ReLAVRA) an increased trend has been observed in carbapenem resistant bacteria since 2014 [26,27]. ReLAVRA is one of the oldest regional networks established by Pan American Health Organization (PAHO) and WHO in 1996 (PAHO, 2021). This network consists of 20 countries, including Argentina, Belize, Bolivia, Brazil, Chile, Colombia, Costa Rica, Cuba, Ecuador, El Salvador, Guatemala, Honduras, México, Nicaragua, Panamá, Paraguay, Perú, the Dominican Republic, Uruguay, and Venezuela. Each country is represented by a national reference laboratory (NRL). AMR data is collected each year to prioritize a number of pathogen-drug combinations [28].

In 2015, right after WHO launched the AMR Global Action Plan, PAHO developed five strategic actions to tackle AMR: (1) raising awareness, (2) surveillance and research, (3) prevention and control, (4) optimizing antibiotic usage, and (5) development of new diagnostics, new antimicrobials, vaccines, and interventions [27].

Patterns of antibiotic consumption have been monitored infrequently in Latin America. The assessment of antibiotic prescription/consumption has been challenging due to poor surveillance systems and unreliable data [29,30]. A study conducted on the utilization of antibiotics in eight Latin American counties from 1997–2007 showed that total antibiotic usage increased in Brazil, Perú, Uruguay and Venezuela. However, in México and Colombia, antibiotic consumption decreased [29]. Recent reports indicate that antibiotic utilization has increased during the past few years in Brazil, México, and Colombia [30–33]. The pharmacologic antibiotic group most commonly consumed varied in different countries but in general beta-lactams (penicillins and cephalosporins), quinolones, macrolides such as azithromycin and trimethoprim/sulfamethoxazole were the most reported [30,33–35]. Consumption and distribution of antibiotics varied greatly in different countries according to population concentration, accessibility and income [29,33,35]. Evaluating antibiotic consumption is challenging and very complex because it involves many variables; there are very few studies, information systems are not reliable and scarce [36]. However, knowledge about the patterns of antibiotic consumption is essential for the development of effective interventions and regulations.

Water is an essential natural resource for all ecosystems and a critical component for human health. However, over the past decades, several factors have dramatically affected the quality of natural (lakes, rivers, marine, groundwater) and artificial (wastewater, drinking water, artificial lakes and ponds) aquatic environments due to human activities [36,37] in Latin America and worldwide. Some of these human activities, such as urbanization, industrialization, farming and animal food production, have led to the dissemination of AMR [38–41]. Water has been recognized as an important vehicle of AMR but the public health impact attributed to the spread of AMR is poorly understood [42].

Some of the major problems favoring the spread of AMR in Latin American countries include great disparities in living conditions, ineffective healthcare systems, unreliable drug suppliers, poor sanitation and sewer disposal, and others [36]. Most of the studies conducted in Latin American countries in aquatic environments were performed in highly anthropogenic impacted environments such as rivers, lakes, and coastal waters which receive wastewater effluents; on the other hand, some of these water sources are used for irrigation and drinking water production. In these studies it is common to find phenotypic identification of resistant bacteria, ARG and mobile genetic elements reported [40,42,43].

The aim of this review is to analyze the anthropogenic activities that introduce MDRO to various aquatic environments in Latin America, as well as to investigate how these water bodies were utilized in different human activities such as drinking water production, vegetable irrigation, and aquaculture, among others. Here we present a summary of our findings and detailed information about the sources of resistant bacteria, identification procedures and antimicrobial resistant genes in different water sources. This information can be found in the Supplementary Materials.

## 2. Methods

Search Strategy: To understand the effects of human activities in natural and artificial aquatic ecosystems in Latin American countries we reviewed the most relevant studies related to antibiotic resistance, antibiotic use and how these water bodies were employed in communities or in populations. To include the maximum quantity of information a Google Scholar search was conducted in two languages, Spanish and English; in Spanish the words "bacterias resistentes" and "de aguas" was used, in the 2000–2020 timeframe, and 4560 entries were identified. All entries were checked by reviewing the title and abstract; only works published in indexed journals focusing on water ecosystems were included. Reviews published in Spanish were not included. For research studies published in English, in addition to Google Scholar, we searched additional databases including Science Direct and PubMed. A total of 23 Latin American countries were included: North America (México), Central America (Belize, Guatemala, Honduras, Nicaragua, El Salvador, Costa Rica, Panamá), South America (Argentina, Bolivia, Brazil, Chile, Colombia, Ecuador,

French Guiana, Guyana, Paraguay, Perú, Suriname, Uruguay, Venezuela) and the Caribbean (Cuba, Dominican Republic, Haití). We configured the search as follows: "water" AND "resistant bacteria" AND "México" OR "Guatemala" OR "Honduras" OR "El Salvador" OR "Nicaragua" OR "Costa Rica" OR "Panamá" OR "Colombia" OR "Venezuela" OR "Peru" OR "Ecuador" OR "Bolivia" OR "Brazil" OR "Argentina" OR "Chile" OR "Uruguay" OR "Paraguay" OR "Cuba" OR "Haiti" OR "Dominican Republic" OR "Belize", OR "Suriname" OR "French Guiana" OR "Guyana", within the same timeframe (2000–2020). A total of 17,700 entries were recovered; the majority were either not related to antimicrobial resistance, water sources or the studies were not conducted in Latin American countries. All entries were checked reviewing the title and abstract. Only works published in indexed journals focusing on water ecosystems were included. Reviews in English were selected to write the introduction. Finally, 72 works were selected, 21 were published research studies in Spanish and 51 in English. Bacteria included Gram negative and Gram positive organisms found in natural and artificial water ecosystems. Susceptibility testing methodology and the level of resistance were evaluated from each report. Publications describing antimicrobial resistant determinants such as integrons and antimicrobial resistance genes were included.

Data Extraction: this was a descriptive study. Data obtained from included publications were exported to a database using a custom-made form. The database included:

a.   Water sources, where the studied water samples were obtained, comprising rivers, and irrigation channels (surface water), seawater, and wastewater (treated and untreated).
b.   Country where the studies were performed.
c.   Anthropogenic activity affecting water bodies: agricultural activities, drinking water production, and wastewater discharges.
d.   Reported antibiotic susceptibility analysis and antibiotic genetic marker detection methods.
e.   Antibiotic resistance levels/Antibiotic resistance genes presence.

All these data are presented in the Supplementary Materials (Tables S1 and S2).

Data Analysis: We predicted that countries with high gross domestic product (GDP) were able to conduct more research studies; for this we ran a Spearman correlation between the number of published studies and the 2020 GDP [44]. The frequency of the most common antibiotic group types reported was evaluated. Additionally, we categorized the studies into two regions: Region 1 included North Latin America, Central America, and the Caribbean (NCC), and Region 2 included South America (SA). Statistical analysis and graphics were done using R program [45].

## 3. Results

A total of 72 publications were selected to review the studies conducted in Latin American countries on various aquatic environments, including natural and artificial water sources, in relation to antimicrobial resistance and its presence in human activities. Brazil was the country that published the most studies (31) [41,46–74], followed by México (10) [37,75–83], Argentina (7) [84–90], Colombia (5) [91–94], Cuba (4) [95–98], Ecuador (4) [99–102], Chile (3) [103–105], Venezuela (3) [106–108], Costa Rica (2) [109,110], Bolivia (1) [111], Nicaragua (1) [112], and Perú (1) [113]. Figure 1 illustrates the number of published studies conducted by country. As we predicted, the Spearman correlation between the number of publications and the GDP was 0.90 ($p = 0.0003593$), which indicates that the more resources the country has, the more research is conducted.

Water sources studied in Latin America included: sea, surface, drinking, wastewater, irrigation, groundwater, estuarine, thermal and recreational waters as is shown in Supplementary Materials. In Figure 2 we present a summary of the water sources studied. Surface water was the most commonly studied (rivers and streams) (30), followed by wastewater (11), sea water (coastal water and recreational waters) (8), multiple water sources (for example river water and drinking water; recreational and irrigation water) (7), drinking water (6) lake water (3), and other, which included estuarine, thermal, groundwater and different recreational waters (7).

The anthropogenic activities related to water sources were reviewed to evaluate how different water bodies were utilized in different countries. We found very diverse usage of water sources by country. For example, 21 studies provided information on water bodies receiving treated/untreated wastewater; there were 11 reports on drinking water production; 10 studies reported on agricultural activities; hospital wastewater was discussed in 7 papers; finally, there were 14 studies reporting multiple human activities involved in a given water source, for example a lake that is used for irrigation, and also for drinking water production and recreational activities; other studies related to human activities (9) included studies with a great diversity of uses such as recreational, thermal waters, groundwater, and others. (Figure 3A).

In order to assess the work done according to research priorities per country, we categorized the studies into two regions: Region 1 included North Latin America, Central America and the Caribbean (NCC), and Region 2 included South America (SA). To avoid bias in the comparison of studies done per country we decided to use percentages (since Brazil had the highest number of studies in Latin America) and divided them by region. As shown in Figure 3B, 46% of water bodies receiving treated/untreated wastewater were reported by region 1, while 24% were reported by region 2; region 1 reported 18% for multiple human activities involving a given water source, while region 2 reported 20%; for drinking water production, region 1 reported 6% and region 2 18%; 18% of agricultural activities were reported by region 1 and 13% by region 2; studies on hospital wastewater were 6% in region 1 and 11% in region 2; 6% of other activities were reported from region 1 and 14% in region 2.

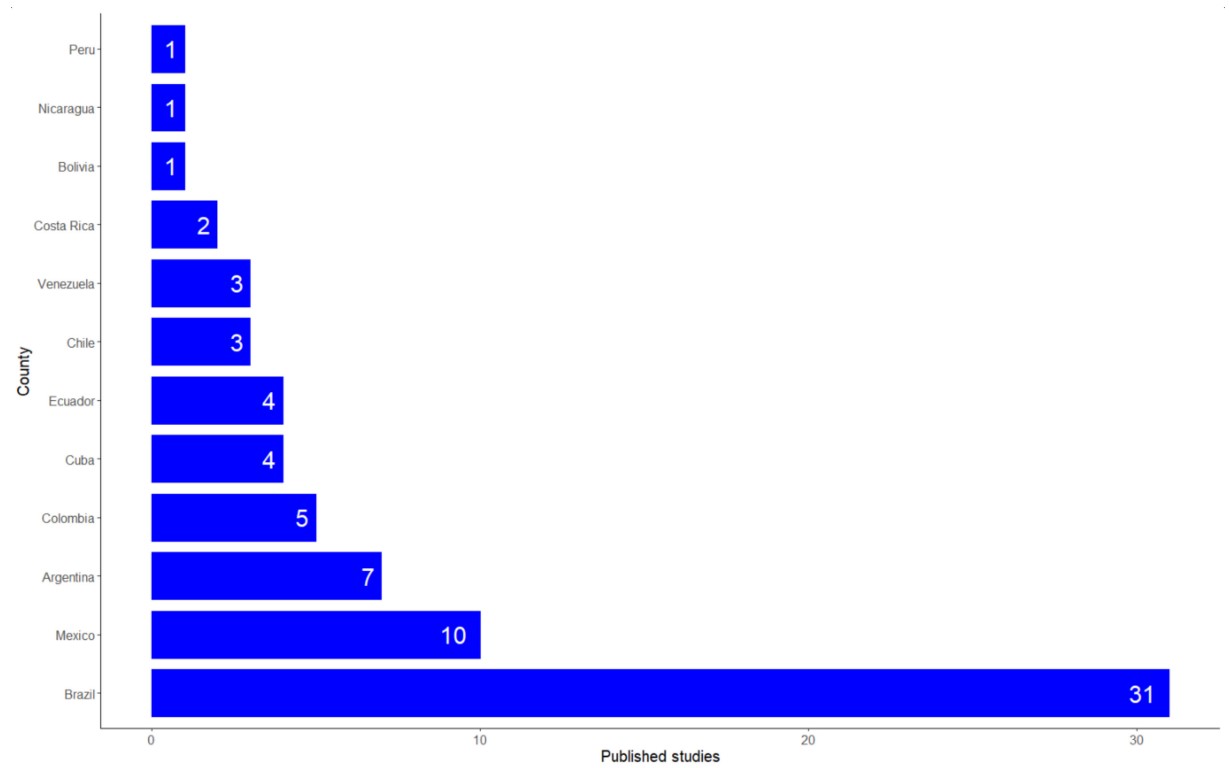

**Figure 1.** Latin American countries studying aquatic ecosystems. Brazil is the leading country with 31 publications, followed by México 10, Argentina, 7 Colombia 5, Cuba and Ecuador 4, Chile and Venezuela 3, Costa Rica 2 and Perú, Nicaragua and Bolivia 1.

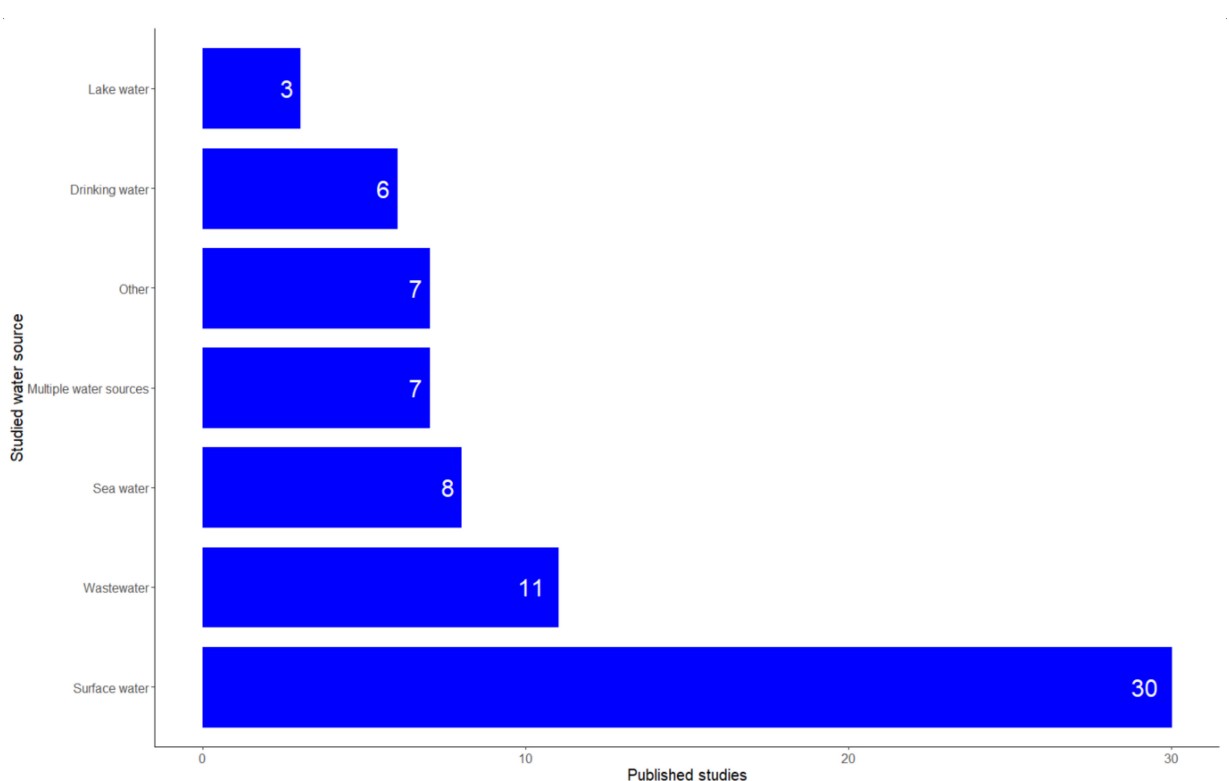

**Figure 2.** Most common water sources studied in Latin American countries. Multiple water sources included: river and drinking water, recreational, and irrigation water. Other water sources included estuarine, thermal, groundwater and different recreational waters.

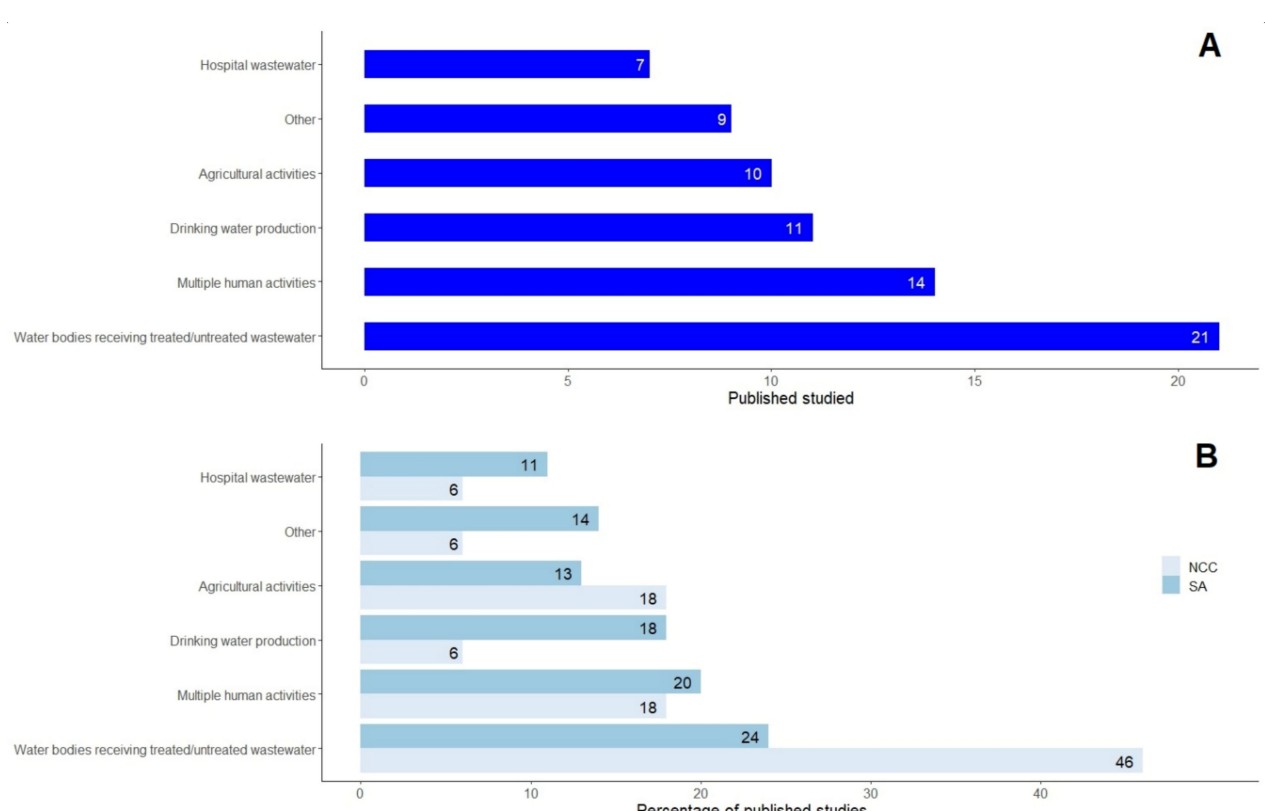

**Figure 3.** Anthropogenic activities affecting water bodies. (**A**) Number of articles published on water use. (**B**) Percent of publications by region. Region 1: North America, Central America and the Caribbean (NCC); Region 2: South America (SA).

Bacterial antimicrobial resistance was studied phenotypically and genotypically. From the 72 research studies reviewed, 59 provided information on bacterial phenotypes. The most common antibiotics tested were cephalosporins, followed by aminoglycosides and penicillin. A summary of the different antibiotic groups tested is shown in Table 1. This Table also presents the percentage of studies that found resistant microorganisms in the aquatic environments in Region 1 (NCC) and Region 2 (SA). It is important to note that all the studies reported from NCC and SA found resistant microorganisms to carboxypenicillins, first generation cephalosporines, macrolides, and nitrofurans, since this groups of antibiotics have been used for a long time. Microorganisms resistant to carbapenems were more prevalent in SA than NCC. The most common technique used to assess antimicrobial susceptibility was the Kirby–Bauer disk diffusion method; other techniques used were the micro-broth dilution by automated analyzers such as Vitek® (bioMérieux, Inc., Marcy-l'Ètoile, France). Information about antimicrobial resistant genes was found in 27 publications. Of these, 23 reported the presence of ESBL genes, 7 reported quinolone resistant genes, 3 reported the presence of integrons and 12 reported the presence of other resistance genes. Most of these studies were done in bacterial isolates; a few studies were conducted using genomic DNA. Detailed information is presented in Table S2.

**Table 1.** Percentage of studies that reported antimicrobial resistance in aquatic environments, by region and antibiotic class.

| Antibiotic Group | NCC | SA |
| --- | --- | --- |
| Aminoglucoside | 93 | 89 |
| Aminocumarine | 100 | NT |
| Amphenichol | 92 | 83 |
| Ansamycin | NT | 100 |
| Carbapenem | 40 | 84 |
| Carboxipenicillin | 100 | 100 |
| Cephalosporin (1st generation) | 100 | 100 |
| Cephalosporin (2nd generation) | 100 | 90 |
| Cephalosporin (3rd generation) | 79 | 96 |
| Cephalosporin (4th generation) | NT | 100 |
| Fluoroquinolone | 100 | 83 |
| Glycopeptide | 100 | 71 |
| Glycylcycline | NT | 75 |
| Inhibition nucleic acid synthesis | 100 | 87 |
| Lincomycin | NT | 100 |
| Macrolide | 100 | 100 |
| Nitrofurane | 100 | 100 |
| Penicillin | 100 | 94 |
| Penicillin + betalactamase inhibitor | 80 | 90 |
| Phosphonic | NT | 100 |
| Polymyxin | NT | 75 |
| Quinolone | 87.5 | 83 |
| Tetracycline | 100 | 97 |

NT: non-tested.

## 4. Discussion

There was a significant difference among Latin American countries' publications. This may be related to the gross domestic product (GDP) of each country ($p$ = 0.0003593). According to the International Monetary Fund, Brazil is listed at the top for GDP followed by Mexico and Argentina [114]. This correlates very well with the three countries that contributed the most publications. However, it could be possible that not all countries are interested in studying AMR or researching water ecosystems. No research studies were found for the following countries: Guatemala, Honduras, El Salvador, Panamá, Paraguay, Haiti, Dominican Republic, Belize, Suriname, Uruguay, French Guiana and Guyana. These findings are in agreement with those reported by Moreno-Switt (2020).

The most common studied water type was surface water. This water ecosystem has a wide variety of uses, which include irrigation, drinking water production, recreational activities and receivers of wastewaters. One big issue to be solved in Latin America is fecal contamination management. Only 37% of the LA countries have population access to improved sanitation facilities [115]. Another great concern is the great disparity in living conditions of communities in different countries. Rural communities that do not have sanitation facilities could disseminate MDRO by fecal material release to the environment; unfortunately, data about these populations is not available [115]. In many countries the availability of potable water data is unknown [115].

The second water source most commonly studied was wastewater. These ecosystems are of great concern since reports indicate that wastewaters are hot spots for AMR [42,116]. In addition, hospital wastewater effluents may play a role in dissemination of MDRO and ARG into the community and environment. For example, Spindler et al., 2012, reported the dissemination of carbapenem, cephalosporins resistant bacteria and the presence of integrons [58]. Resistance to several antibiotics (ESBL, 3rd generation cepholosporins, tetracyclines, quinolones and carbapenems) were reported by Picão et al., 2013 [60]. Wastewaters were studied intensely in South America [50,55,57,72]. Irrigation water has also been linked to the dissemination of AMR [82,84]. Contaminated irrigation water used in crop production has been of great concern since contaminated vegetables may lead to infections or other health consequences. Currently, the WHO Guidelines for Wastewater Use and Reuse do not include control of MDROs or antibiotic detection [117]. It is important to mention that only 50–60% of Latin America is connected to sewerage, and only 30% of domestic sewage is treated [118]. Due to the lack of financial resources in most countries, the management of wastewater is difficult.

Despite the significant threat that AMR represents worldwide and the urgent call for global action, many Latin American countries have not fully enforced policies to prevent the spread of AMR. Laws prohibiting over-the-counter antibiotic sales have been in place in several countries since 2010 [36,119]. However, effective surveillance programs and data regarding the impact on the decrease of antibiotic consumption are still not available [120]. In the veterinary and food production animal sectors, data on therapeutic and prophylactic use of antibiotics are still unavailable [121,122]. Chile, which is one of the countries leading the control of AMR in South America, has not yet established its National Plan to tackle AMR [123]. In summary, antibiotic use and AMR have been low on Latin American governments' policy agendas. More comprehensive strategies, such as antibiotic stewardship programs, improvement of prescription quality, public health awareness and assessment of restrictive and structural interventions need to be instituted and enforced.

The interest and complexity of AMR research studies conducted in Latin American countries has increased during the last few years. At first, most of the AMR studies identified bacterial isolates phenotypically and tested antimicrobial susceptibility by diffusion methodology. However, most recent studies included advanced technologies such as automated analyzers, Vitek® and MALDI-TOF, and molecular identification by PCR [54,70,102]. Bacteria in these studies included a wide variety of genera and species. Gram negative bacilli were the most studied, specifically the *Enterobacteriacea* family and particularly *E. coli* and some non-fermenter bacteria such as *Pseudomonas* spp. and *Acinetobacter* spp. Supplementary Materials presents detailed information on the bacteria studied.

## 5. Conclusions

Despite the low number of reports on AMR and anthropogenic activities impacting water ecosystems in Latin American countries, information obtained in this study indicates that the spread of AMR is overwhelming. Latin American countries have diverse socioeconomic differences, and significant disparities exist in living conditions and lack of financial resources. Several factors contribute to the spread of AMR, such as lack of infrastructure and education, easy access to antibiotics and a lack of regulations. Development of public

policies, public health awareness, better diagnostics and international surveillance may contribute to alleviating the serious AMR crisis.

**Supplementary Materials:** The following are available online at https://www.mdpi.com/article/10 .3390/w13192693/s1, Table S1: Phenotypic evaluation of antimicrobial resistant bacteria collected from various aquatic environments in Latin America; Table S2: Antimicrobial Resistant Genes and mobile genetic elements reported from Latin American countries.

**Author Contributions:** Conceptualization and designed D.C.D. and L.M.C.; administration and coordination, D.C.D.; data analysis and interpretation D.C.D. and L.M.C., writing, editing D.C.D. and L.M.C.; literature review D.C.D., L.M.C.; L.M.C. and D.W., tables; L.M.C., D.C.D. references; L.M.C., figures. All authors have read and agreed to the published version of the manuscript.

**Funding:** This project was supported in part by the National Institute of General Medical Sciences of the National Institutes of Health under award numbers R25GM123928. The content is solely the responsibility of the authors and does not necessarily represent the official views of the National Institutes of Health.

**Institutional Review Board Statement:** Not applicable.

**Informed Consent Statement:** Not applicable.

**Data Availability Statement:** Not applicable.

**Conflicts of Interest:** The authors declare no conflict of interest.

## Abbreviations

The following abbreviations are used in this manuscript

| | |
|---|---|
| AMR | Antimicrobial Resistance |
| ARG | Antimicrobial Resistant Genes |
| CDC | Centers for Disease Control and Prevention |
| ESBL | Extended Spectrum Beta Lactamases |
| GARP | Global Antibiotic Resistance Partnership |
| GDP | Gross Domestic Product |
| GHSA | Global Health Security Agenda |
| LMIC | Low- and Middle- Income Countries |
| LIC | Low- Income Countries |
| MALDI-TOF | Laser Desorption Ion Matrix Assistedization Time of Flight |
| MIC | Middle- Income Countries |
| MIC | Minimum Inhibitory concentration |
| MRDO | Multidrug Resistant Organisms |
| PAHO | Pan American Health Organization |
| PCR | Polymerase Chain Reaction |
| TATFAR | Transatlantic Taskforce on Antimicrobial Resistance |
| WHO | World Health Organization |
| WWTP | Wastewater Treatment Plants |

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
