# Peer review of "Anthropogenic Activities and the Problem of Antibiotic Resistance in Latin America: A Water Issue"

_water, doi:10.3390/w13192693_

Round 1

Reviewer 1 Report

The study is interesting and, above all, alerts to a topic that everyone should pay attention to in order to avoid future problems in terms of public health. Nonetheless, I believe that the methodology used to carry out the analysis of the collected data (merely descriptive) is a below what is expected in an article of this nature. With the collected data, I believe that the analysis methodology could be much more interesting, enriching the material that can be discussed. So much so that the authors begin their discussion saying "There was a significant difference among Latin American countries publications". However, this cannot be said, as no statistical analysis was performed to support this significant difference. This kind of statement happens later, as in "This correlates very well with the three countries that contributed to most publications", although there is no methodology to study the correlation.

In this sense, I reinforce that the study is interesting, but the data analysis methodology must be improved, in order to support the results achieved.

Author Response

Reviewer 1

Thank you for your well-argued suggestions, which improved our manuscript. Indeed, the reviewer is correct in that the statements "There was a significant difference among Latin American countries publications" and "This correlates very well with the three countries that contributed to most publications", cannot be made without running a statistical test. We modify the methods section related to data extraction and data analysis. We supported our statements by running a Spearman correlation between the number of published studies and the 2020 GDP of selected Latin American countries. We have also made a major revisions throughout our manuscript to clarify its content and added more in depth information in several paragraphs. The supplementary material has been re-organized according to country and in alphabetical order.

Reviewer 2 Report

General comments: This manuscript highlights the research and reports that have been carried out in Latin America on antibiotic resistance and the prevalence of ARG and ARB in water systems. This information is always important for understand how widespread AMR is all over the world and what factors are driving the increase in prevalence. The paper contains lots of relevant information and will help those in these countries to better determine their course of action if they wish to reduce the impact of AMR in the future. However, the manuscript lacks depth in the discussion around WHY certain antibiotics, or water course or country were targeted. The number of publications should not be the sole criteria used to classify importance or relevance. These 72 articles contain a lot more information that could help articulate the argument the authors are trying to make: that being that anthropogenic activities and lack of regulations promote more AMR in the aquatic environment. Furthermore, there is no discussion about legislation, current or proposed? The abstract suggests that there is a lack of legislation yet the authors do not substantiate that point with research and do not suggest what kinds a guidelines/policies are needed to reduce the threat of AMR? Specific comments: Line 51-55: need two sentences here. Last part of current sentence is a new thought. Line 71: remove capital on Low in first sentence, is OK in second sentence Line 91: write out PAHO as it is the first time it appears in manuscript Line 139: finish sentence with “in two languages: Spanish and English”. Then start new sentence with “ In Spanish….” Line 133-136: This paragraph is not consistent with the review, the review does not look at the impact of AMR on various aqueous system but instead how many times a water system have been identified to contain AMR. More papers do not necessarily mean more impact? Same for all of the properties that were looked at. Needs to be re-worded. Line 142: focusing ‘on’ water ecosystems… Line 156: is there a reference to back up “same criteria as previously described”? Were review articles in English also eliminated? Overall, the search criteria is not clear… Line 172-175: formatting not consistent between countries, please check Line 196-197: Descriptive terms use in dialogue are different than those used in table, please be consistent with terminology Line 199-206: Table categorizes regions by NCC and SA and paragraph uses region 1 and 2, please be consistent otherwise readers can not follow argument. Line 229: what property are you saying has a significant difference? Number of publications? Resistances? Types of antibiotics? Research? Not clear and no statistics done so not sure how the authors can say there was a significantly different. Line 243: how does lack of information suggest rural communities are at risk? I don’t see the connection? Line 245: new paragraph for wastewater If wastewater is often studied, and are found to be hotspots what does that have to do with the lack of legislation? The authors suggest that the management of wastewater is difficult to control, why? where is that data? There is no discussion of legislation of water or wastewater or AMR in any country? What type of legislation do the authors propose? How easy or difficult will that be to do? List of abbreviations: check formatting, not aligned in my version but could be just the download. Appendix: Table 1 Please list information in table using some sort of category (ie. either List by country alphabetically or list by water type alphabetically or by year or something……) the reader needs to be able to find info quickly and hard to do so when there is no rational on how the information is listed.

Author Response

Reviewer 2

General Comments

We are thankful to the reviewer who made cogent and helpful suggestions for the improvement of the manuscript.

We have modified the methodology and the results sections so that the discussion will be more meaningful. We have also added a paragraph in the discussion in regard to legislation (Line 301-313).

The antibiotics were not selected, we have described all the antimicrobials that were reported in the studies. Therefore we eliminated Figure 4 and a new table (Table 1) was created indicating the percentage of studies that found resistance to a particular antibiotic by region.

The water sources were obtained directly from the research studies. To better explain the data extracted we added a paragraph (Line 171-181) in the method section.

We did not selected the countries. We listed the countries found in the research papers within the 72 publications.

We have modified extensively the manuscript to convey the information more clearly.

The supplementary material has been re-organized according to country and in alphabetical order.

Specific Comments

New             Revised

Line 72        (71): the capital on Low was changed for low

Line 91        (91): PAHO was spelled out Pan American Health Association

Line 145     (139): the sentence was finished with the words Spanish and English

New                                         Revised

Line 133-138                           (133-136): This paragraph has been reworded

Line 148                                  (142): “on” has been added after focusing

Line 163-165 and 171-189      (156): Reviews in English were included. The methodology

                                                           sectionwas modify to clarify its content

Line 193-198                         (172-175): Formatting of countries and respective number of

                                                             studies have been corrected

Line 205-212                          (196-197): Terminology of text and table have been matched

Line 228-230                          (199-206): The region 1 (NCC) and region 2 (SA) have been changed in text so they will correlate with the Figure

Line 269                      (229): We have run a Spearman correlation test to justify the statement “There was a significant difference among Latin American countries publications”. The gross domestic product (GDP) correlated well with the number of publications per country (p=0.0003593). Also in the results section we added a brief paragraph to explain the statement (198-200)

Line 282-284             (243):  An explanation have been added 

Line 296-300              (245):  A new paragraph for wastewater have been created. We made modifications for clarification.

Line 301-313              New paragraph on policy/legislation

Round 2

Reviewer 1 Report

The authors responded in accordance with what was requested, so I believe it can go on for publication.